

# Novel inflammatory cell infiltration scoring system to investigate healthy and footrot affected ovine interdigital skin

Michael Agbaje[*], Catrin S. Rutland[*], Grazieli Maboni, Adam Blanchard, Melissa Bexon, Ceri Stewart, Michael A. Jones and Sabine Totemeyer

School of Veterinary Medicine and Science, University of Nottingham, Nottingham, United Kingdom
[*] These authors contributed equally to this work.

## ABSTRACT

Ovine footrot is a degenerative disease of sheep feet leading to the separation of hoof-horn from the underlying skin and lameness. This study quantitatively examined histological features of the ovine interdigital skin as well as their relationship with pro-inflammatory cytokine (IL-1β) and virulent *Dichelobacter nodosus* in footrot. From 55 healthy and 30 footrot ovine feet, parallel biopsies (one fixed for histology) were collected post-slaughter and analysed for lesions and histopathological analysis using haematoxylin and eosin and Periodic Acid-Schiff. Histological lesions were similar in both conditions while inflammatory scores mirror IL-1β expression levels. Increased inflammatory score corresponded with high virulent *D. nodosus* load and was significant ($p < 0.0001$) in footrot feet with an inflammatory score of 3 compared to scores 1 and 2. In addition, in contrast to healthy tissues, localisation of eubacterial load extended beyond follicular depths in footrot samples. The novel inflammatory cell infiltration scoring system in this study may be used to grade inflammatory response in the ovine feet and demonstrated an association between severity of inflammatory response and increased virulent *D. nodosus* load.

Corresponding author
Sabine Totemeyer,
sabine.totemeyer@nottingham.ac.uk

## INTRODUCTION

Lameness in sheep, caused by interdigital dermatitis (ID) and footrot, is widespread in the UK, having a major welfare and economic impact. Footrot is caused by *Dichelobacter nodosus* which occurs in both virulent and benign forms. Footrot is defined by separation of the hoof from the underlying structures and an accumulation of necrotic material, with varying degrees of severity (*Beveridge, 1941*; *Thomas, 1962*), this damage is believed to be mediated by immune pathology rather than by bacterial enzymes and toxins (*Egerton, Roberts & Parsonso, 1969*).

Footrot development is characterised by invasion of neutrophils and lymphocytes into the dermis and epidermis in response to bacterial invasion of the epidermis (*Davenport et al., 2014*; *Egerton, Roberts & Parsonso, 1969*). Early histological studies described footrot as a degenerative condition of the stratum granulosum and spinosum which results in cellular

degeneration (cell ballooning), formation of micro-abscesses and vacuoles which coalesce and progress to cavities (*Beveridge, 1941*; *Deane & Jensen, 1955*; *Thomas, 1962*). A recent histological study of clinically healthy and affected feet showed a progressive increase in lymphocyte and neutrophil infiltration into the dermis and epidermis between healthy, ID and footrot samples (*Davenport et al., 2014*). In footrot samples, purulence was seen in areas of epidermal degeneration, necrosis and epidermal-dermal clefts. As in previous studies, cytoplasmic ballooning and nuclear condensation were observed in the stratum spinosum of the epithelium (*Thomas, 1962*), as well as areas of fibrosis indicating a chronic reaction to tissue damage (*Egerton, Roberts & Parsonso, 1969*).

Recently, we have shown that IL-1β and CXCL-8, but not IL-6 and IL-17 mRNA expression levels correlate with *D. nodosus* load in footrot samples (*Maboni et al., 2017a*). In addition, stimulation of ex vivo organ explant of ovine interdigital skin infected with *D. nodosus* elicited IL-1β release (*Maboni et al., 2017b*).

The aim of this study was to perform a qualitative and quantitative analysis of histopathological features of the ovine interdigital skin comparing healthy and footrot affected feet. In order to evaluate inflammation in the epidermis and dermis a novel scoring system was developed and correlation of those scores with the expression of the pro-inflammatory cytokine IL-1β and *D. nodosus* load was investigated. In addition the depth of within tissue colonisation of eubacteria, *D. nodosus* and *Fusobacterium necrophorum* were examined in the context of hair follicle depth.

## MATERIALS AND METHODS

### Ovine biopsies

Samples of ovine feet were obtained post-slaughter from an abattoir and assessed by two independent scorers for conformation and clinical conditions (healthy and footrot affected).

Conformation scoring was assessing the integrity of the sole and heel/wall of each digit: 0, undamaged sole and heel area with a perfect shape; 1, mildly damaged/misshapen sole and/or heel area of the digit (<25%); 2, moderately damaged/misshapen sole and/or heel area of the digit (>25% and <75%); 3, severely damaged/misshapen sole and/or heel area of the digit (>75%) (*Maboni et al., 2016*). Ovine feet were scored as described previously (*Kaler et al., 2010*) with healthy defined as an absence of any interdigital skin lesion and footrot as the presence of underrunning lesions. Roughage, faeces and mud were removed from the interdigital space of each foot and prior to biopsy taking the skin was wiped with 70% ethanol. Biopsies of 6 mm diameter ($n = 85$) were taken using a punch biopsy tool (National Veterinary Services, UK) in the interdigital space along the skin-hoof interface, of which 55 were from interdigital skin without signs of disease and 30 from feet with footrot.

### Histological sample preparation

Preparation was as previously described (*Davenport et al., 2014*), in brief all samples were fixed using 10% (v/v) neutral buffered formalin; however to obtain optimal tissue integrity healthy biopsies were fixed for 48 h at 4 °C and footrot biopsies for 24 h at room

temperature (fixation and processing were optimised using two fixatives (10% v/v neutral buffered formalin verses 4% v/v paraformaldehyde), two incubation temperatures (room temperature verses 4 °C) and two incubation times (24 h verses 48 h). An extended tissue processing protocol was used (60 min in $dH_2O$, 4 h in 50% ethanol, 4 h in 70% ethanol, 16 h in 90% ethanol, 4 h in 100% ethanol, 4 h in Xylene) followed by embedding into paraffin wax (2 h, 60 °C). Paraffin wax embedded tissues were soaked in 10% (v/v) ammoniated water and 6 μm thick sections were cut from each block by microtome (RM2255; Leica, Wetzlar, Germany). Serial sections were mounted on polysilinated microscope glass slides (Menzel Gläser Polysine®; Thermo-Scientific, Darmstadt, Germany) and dried at room temperature overnight.

## Tissue staining

Paraffin sections were heated at 60 °C for 5–10 min, incubated in xylene twice for 5 min each and rehydrated in 100% ethanol, 90% ethanol, 70% ethanol then twice in $dH_2O$ for 5 min each. Protocols were optimized for haematoxylin and eosin (H&E) and Periodic Acid-Schiff (PAS; Sigma-Aldrich, Dorset, UK). In brief H&E stained samples received 2.5 min in haematoxylin, 15 s in 1% acetic industrial methylated spirits, 15 s in ammoniated water and 4 min in eosin. The PAS stain included immersion in periodic acid-Schiff solution for 15 min, haematoxylin for 3 min. Following each staining protocol sections were dehydrated through an ethanol series.

## Image capture and analysis

Each observer was blinded to the sample identification to avoid subconscious bias. Images were captured using a Leica CTR500 microscope (Leica Microsystems, Darmstadt, Germany) with bright field light. For each sample, three sections approximately 400 μm apart were analysed. At 40× magnification, five non-overlapping photos were taken from each section from both the epidermis (1,275 photomicrographs analysed) and the dermis (1,245 photomicrographs—two fragmented dermal tissue sections were excluded) of H&E stained sections and from the dermal-epidermal junctions of PAS stained sections (1,275 photomicrographs analysed). In addition each sample was viewed at 5× magnification and photomicrographs were merged in order to visualise entire sections.

## Tissue and cell scoring

For each tissue sample, 15 photomicrographs were scored for inflammation. The maximum score of five non-overlapping images from each of three slides per block was determined as the score for that tissue sample. A score of 0 represented no leukocytes in photomicrograph, score 1 occasional infiltration of single leukocytes visible, score 2 focal infiltration of leukocytes, score 3 coalescing leukocytes—individual loci could not be distinguished, and score 4 diffuse infiltration of leukocytes throughout the field of view. Examples of the epidermal and dermal scoring system are shown in Figs. 1 and 2, respectively. The scoring system was validated using two independent scorers blinded to sample identification.

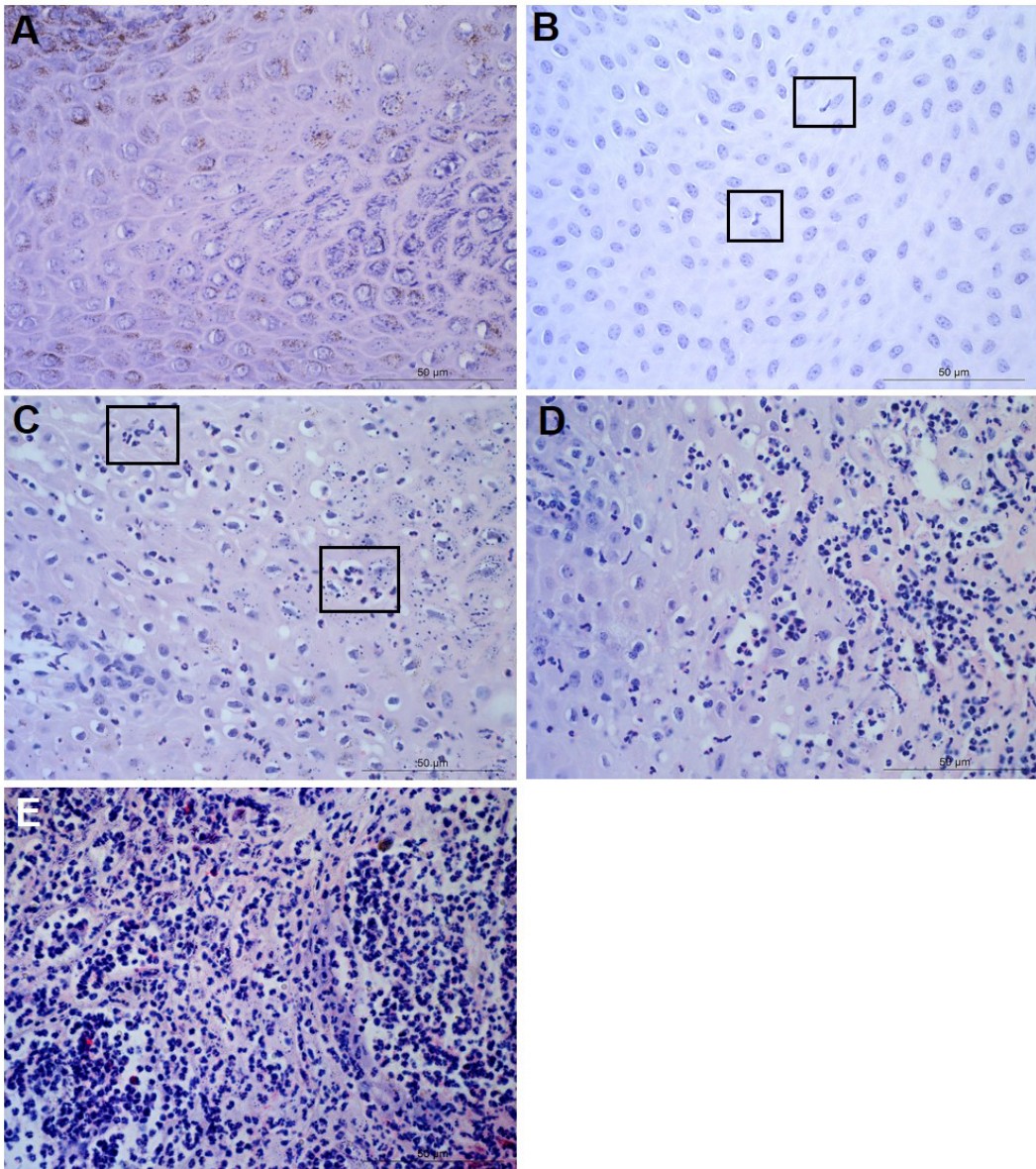

**Figure 1** **Descriptors of leukocyte cell infiltration in ovine interdigital skin epidermis.** Haematoxylin and eosin stained epidermis photomicrographs showing no leukocytes in photomicrograph (A), occasional infiltration—single leukocytes visible within boxed areas (B), focal infiltration of leukocytes—visible in boxed areas (C), coalescing leukocytes—individual loci cannot be distinguished (D) and diffuse infiltration of leukocytes throughout the field of view (E). (A–E) represent inflammatory scores of 0–4 respectively, $n = 55$ healthy and 30 footrot. Scale bars represent 50 μm.

Parakeratosis was defined as the retention of nuclear remnants in the stratum corneum (*Brady, 2004*). Grading in the interdigital skin stratum corneum was based on a three ordinal scale criteria as: the absence of parakeratosis (score 0), the presence of focal parakeratosis (score 1) and diffuse parakeratosis (score 2) (Figs. 3A–3C).

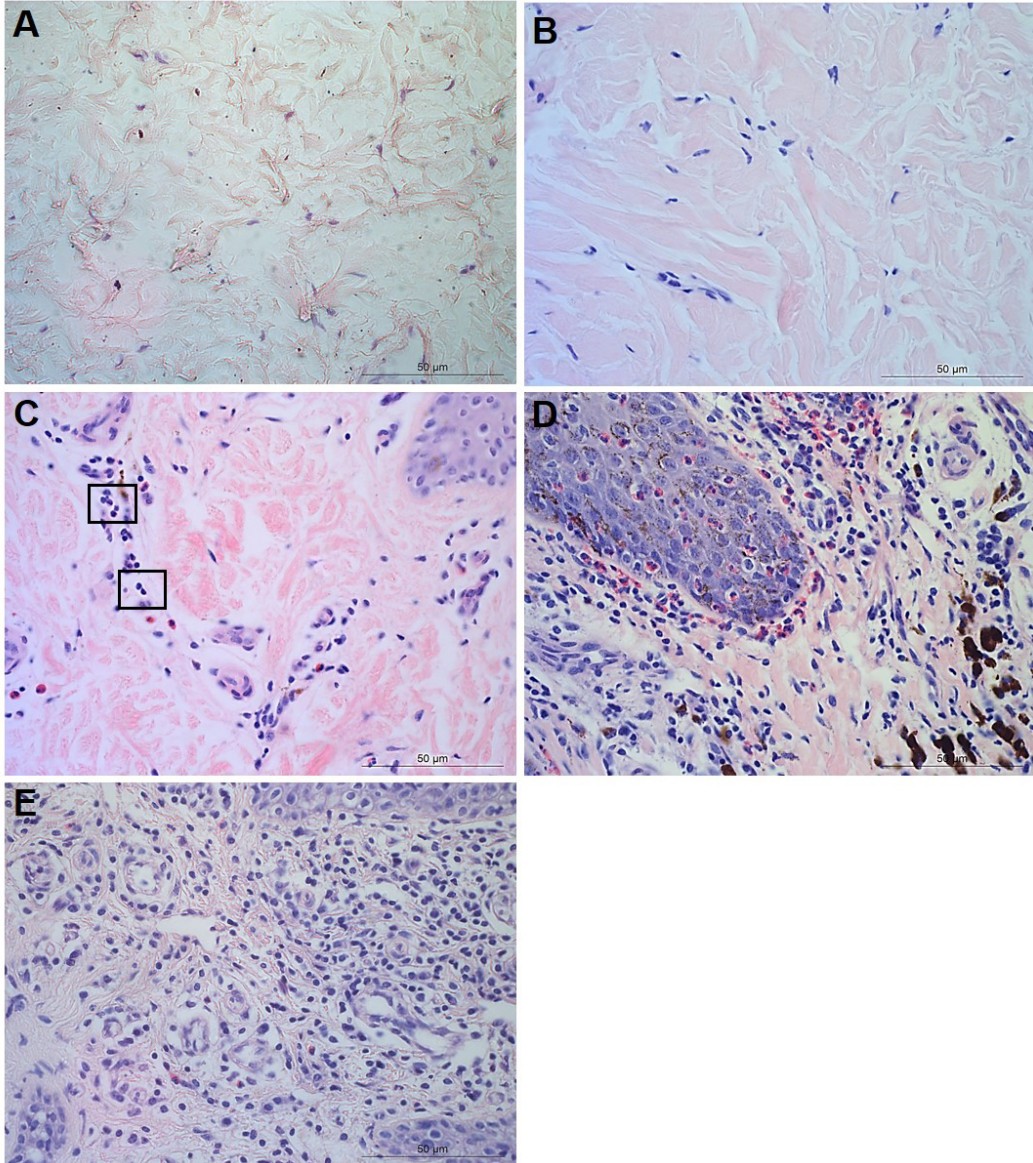

**Figure 2 Descriptors of leukocyte cell infiltration in ovine interdigital skin dermis.** Haematoxylin and eosin stained dermis photomicrographs showing no leukocytes in photomicrograph (A), occasional infiltration of leukocytes (B), focal infiltration of leukocytes—visible within boxed areas (C), coalescing leukocytes—individual loci cannot be distinguished (D) and diffuse infiltration of leukocytes throughout the field of view (E). A–E represent inflammatory scores 0–4 respectively, $n = 54$ healthy and 29 footrot. Scale bars represent 50 µm.

Micro-abscesses were defined as the aggregation of inflammatory cells and cellular debris, confined by fibrotic tissue walling. Grading was according to skin layer (intracorneal, sub-corneal and dermal (Figs. 3D–3F)) and based on a nominal grading scale: presence/absence from five images throughout each tissue section.

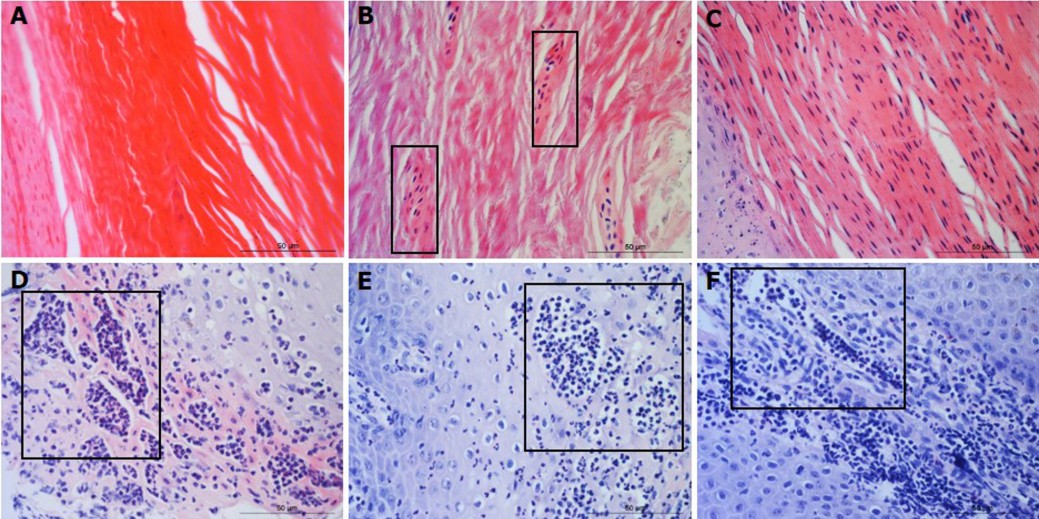

**Figure 3** **Descriptors of parakeratosis and micro-abscesses in ovine interdigital skin.** Haematoxylin and eosin stained interdigital skin from healthy ($n = 55$) and footrot ($n = 30$) affected sheep. Stratum corneum photomicrographs showing no nuclear remnants visible (A), focal accumulation of nuclear remnants (B) and diffuse accumulation of nuclear remnants (C). (A–C) represent parakeratosis scores 0–2 respectively. Representative photomicrographs of micro-abscesses from the intra-corneal (D), sub-corneal (E) and dermal layers (F). Scale bars represent 50 μm.

In order to determine the area of ballooned cells per cross sectional area in the epidermis, photomicrographs were taken, saved in a Tagged Image File Format and uploaded into the analysis software Image Pro 6.3 (Media Cybernetics, Rockville, MD, USA). Each image was calibrated for the measurement of ballooned cells area in absolute values ($\mu m^2$). Ballooned cells were identified and manually selected using the software to trace around each ballooned cell or group of cells. Percentage area of ballooned cells per photomicrograph calculated (total area of ballooned cell(s)/total area of epidermal cross section × 100, for example see Fig. 4A).

Congestion was defined as presence of dilated blood capillaries with visible red blood cells (erythrocytes) within the dermis and haemorrhage was defined as observed blood cells outside capillaries in tissues (Figs. 4B–4C). For both features, 15 photomicrographs were analysed per tissue with a nominal grading scale: presence/absence.

Basement membrane integrity was assessed using PAS stained sections scored for basement membrane disruptions at the dermal-epidermal junction as follows: score 0, no basement membrane disruption identified; score 1, focal basement membrane disruption identified; score 2, multiple basement membrane disruption identified (Figs. 4D–4F).

The IL-1β expression and *D. nodosus* quantification data have been published previously (*Maboni et al., 2017a*).

## Bacterial localisation

Tissue samples were briefly incubated in 70% ethanol followed by overnight incubation in 30% (w/v) sucrose and embedded in OCT (VWR International, Oud-Heverlee, Belgium).

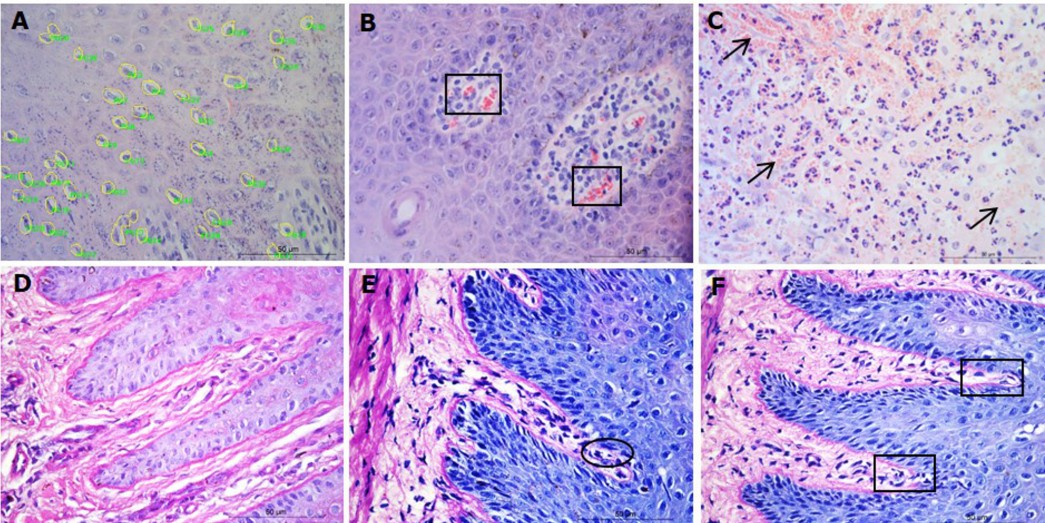

**Figure 4** **Interdigital skin cell ballooning, congestion and basement membrane integrity.** Haematoxylin and eosin stained sections from healthy ($n = 55$) and footrot ($n = 30$) affected sheep showing representative cell balloon measurements (A), congestion (B) and haemorrhage (C). Periodic Acid-Schiff stained tissue shows basement membrane integrity scoring from 0 to 2 (D–F, respectively).

Alternating thick (40 μm) and thin (9 μm) transverse sections were sectioned from the dermal layer across biopsies into the epidermis. The cryostat blade was cleaned with 70% ethanol prior to each thick section in order to prevent potential bacterial contamination. Thick sections intended for DNA extraction to determine bacterial abundance were preserved in 0.5ml RNAlater® (Sigma-Aldrich, St. Louis, MO, USA) at room temperature and incubated overnight prior to DNA extraction. Corresponding alternate thin sections underwent H&E staining. DNA was isolated using the QIAamp cador®kit (QIAGEN, Hilden, Germany) as described previously (*Maboni et al., 2016*). Bacterial load was quantified using quantitative PCR as described previously for total eubacteria (*Strub et al., 2007*), *D. nodosus* (*Frosth et al., 2012*) and *F. necrophorum* (*Frosth et al., 2015*). In order to compare bacterial localisation and load across the same depth of skin layers across different samples, 11 bin groups representing skin depths from 0 to 2,200 μm (range 200 μm/bin) were created. $N = 4$ healthy and five footrot samples.

## Statistical analysis

Statistical analyses were performed on Graphpad Prism version 6 for Windows. Resulting data were presented as frequencies and percentages. Categorical data within and between clinical conditions were compared by Fisher's exact and Chi-square tests while continuous data were analysed by Student $T$-test or Kruskal Wallis test, dependant on data distribution. Statistical data on DNA samples was carried out using Pearson correlation. Analysis was taken as significant when $p \leq 0.05$.

## Ethical approval

Ethical approval was obtained from the School of Veterinary Medicine and Science Ethics committee, University of Nottingham (ethical approval number: 796 130128).

## RESULTS

To investigate the levels of inflammation within the interdigital skin, inflammation and occurrence of micro-abscesses were scored separately in the epidermis and dermis. In addition, in the epidermis, the area of ballooning was measured and the occurrence of parakeratosis was determined, and in the dermis, occurrence of haemorrhages and congestion were determined. The basement membranes were also assessed for disruptions.

In the epidermis, the same level of inflammation was observed in healthy ($n = 55$) and footrot ($n = 30$) samples with a median inflammation score 2 (minimum 1, maximum 4, Fig. 5A). In addition, no differences between healthy and footrot samples were seen with regards to the area of ballooned cells (Fig. 5B; $n = 55$ and 30 respectively). The proportion of samples with intra-corneal (healthy 43.64%, 24/55 (95% CI [59–90.4%])) and footrot 33.33% 10/30 (95% CI [27.2–73%])) or sub-corneal (healthy 25.45%, 14/55 (95% CI [20–50%])) and footrot 23.33%, 7/30 (95% CI [13.2–53%]) micro-abscesses were also not significantly different (Figs. 5C–5D). Parakeratosis was also observed to a very similar extent in both, healthy (72.73%, 40/55 (95% CI [59–84%])) and footrot (76.67%, 23/30 (95% CI [58–90%])) samples (Fig. 5E). However, while in footrot these were evenly split into samples with diffuse and focal parakeratosis, two thirds of the healthy samples with parakeratosis showed a diffuse pattern (Fig. 5E).

Basal membrane disruption was observed in 74.55% (41/55 (95% CI [61–85.3%])) healthy and 56.67% (17/30 (95% CI [37.4–74.5%])) footrot affected samples, with only a small proportion of samples with multiple disruptions (Fig. 5F).

In the dermis, a higher maximum inflammation score of 3 (minimum 2, maximum 4) was seen in healthy and footrot affected tissues (Fig. 6A; $n = 54$ and 29 respectively). There was also no difference between healthy and footrot affected tissues with regards to the proportion of samples with dermal micro-abscesses, haemorrhages or congestion (Figs. 6B–6D). Dermal micro-abscesses were observed in 8.5% of the tissues, with four out of 50 healthy tissues (7.41% (95% CI [2.2–19.2%])) and three out of 26 footrot affected tissues (10.34% (95% CI [2.4–30.1%]); Fig. 6B). Haemorrhages were observed in approximately one third of samples of each disease state (Fig. 6C; healthy = 30.91%, 17/50 (95% CI [28.6–62%]), footrot = 30.00%, 9/30 (95% CI [22–66%])). A similar level of congestion in blood vessels was found in clinically healthy (49.00%, 27/55 (95% CI [81.6–99.9%])) and footrot samples (40.00%, 12/30 (95% CI [41–86.6%])) (Fig. 6D).

We have shown previously that mRNA expression of IL-1β, similar to inflammation scores, were comparable in healthy and footrot affected tissues in parallel samples from the same feet (*Maboni et al., 2017a*). Here we analysed inflammatory scores, parakeratosis and presence of micro-abscesses in the context of IL-1β mRNA expression. IL-1β expression was significantly higher in the presence of diffuse parakeratosis in healthy tissues with no difference in footrot affected tissues (Figs. 7A–7B; $n = 41$ healthy and 23 footrot, $p < 0.01$).

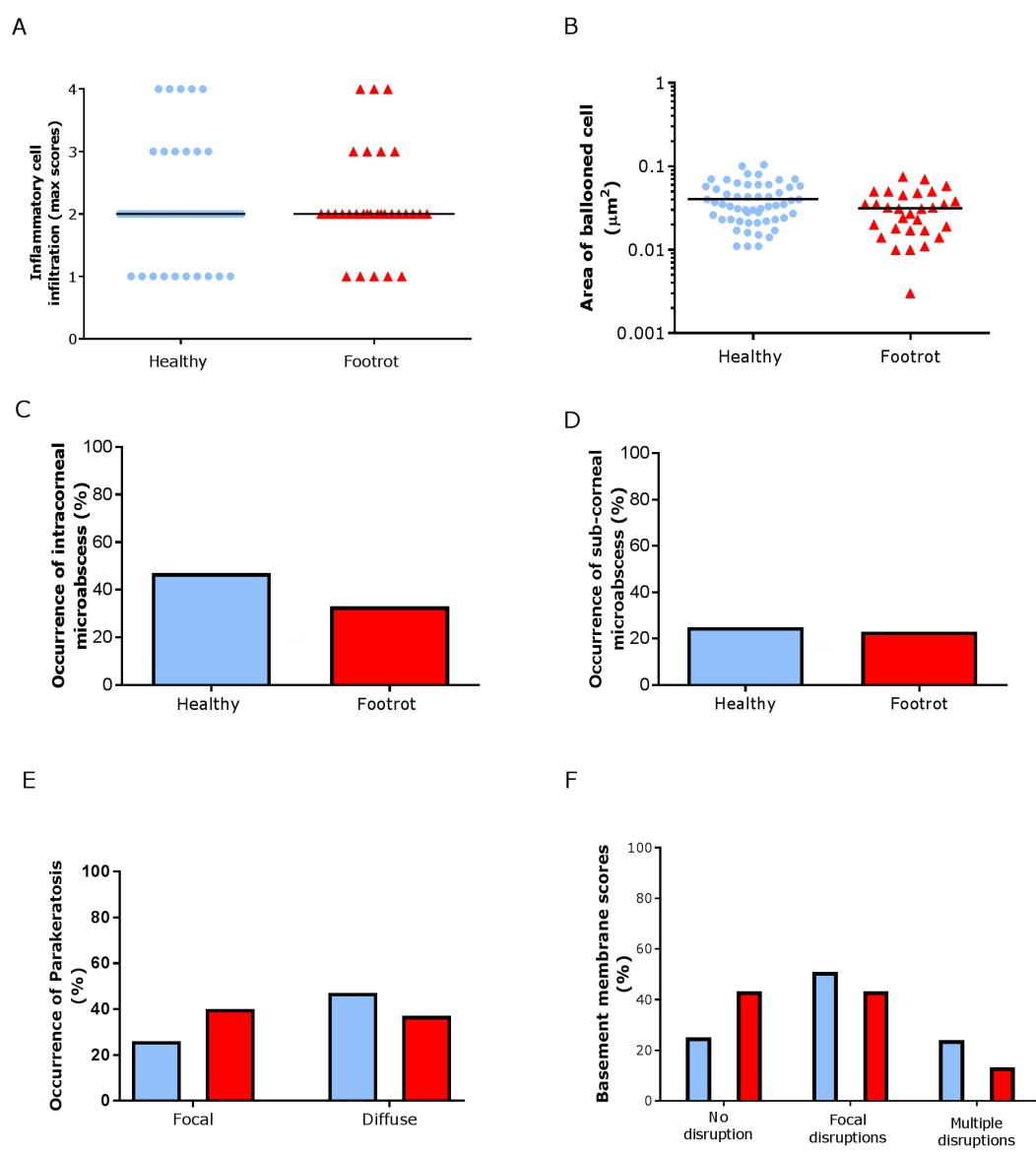

**Figure 5** **Epidermal histological lesions of ovine interdigital skin basal membrane integrity.** Haematoxylin and eosin stained sections from 55 clinically healthy and 30 footrot tissue samples of skin/hoof interface were evaluated with regards to inflammatory score (A), area of cell ballooning (B), presence of intra-corneal micro-abscesses (C), presence of sub-corneal micro-abscess (D), and parakeratosis score (E). PAS stained sections from the same tissues were scored for disruptions of the basement membrane (F). The horizontal black line indicates median (A) and mean (B) values. Statistical analysis: Fisher's exact test (A, C–E) and one-way ANOVA (B).

When comparing relative IL-1β expression with the inflammation scores they mirrored to an extent (Fig. 8).

Since footrot pathology is mediated by host inflammatory responses whilst the disease is initiated by *D. nodosus*, a virulent *D. nodosus* load against host inflammatory cells was compared in healthy ($n = 30$ for both epidermis and dermis) and affected tissues ($n = 18$

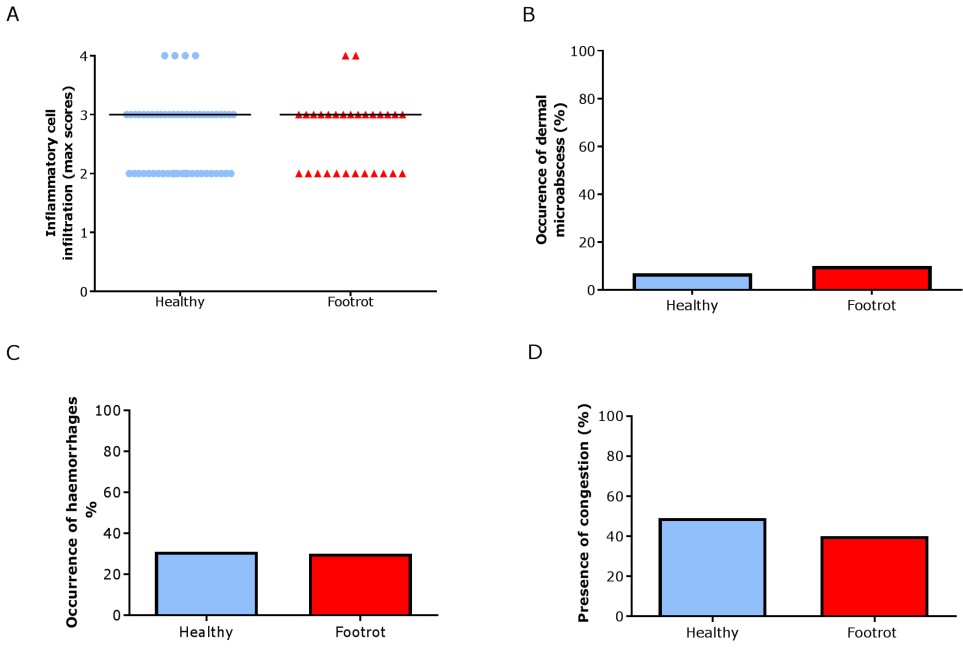

**Figure 6  Dermal histological lesions of ovine interdigital skin.** Haematoxylin and eosin stained sections from 55 clinically healthy and 30 footrot tissue samples of skin/hoof interface were evaluated with regards to inflammatory score (A), presence of dermal micro-abscesses (B), haemorrhages (C), and congested blood vessels (D). The horizontal black line indicates median (A) values. Statistical analysis: Fisher's exact test.

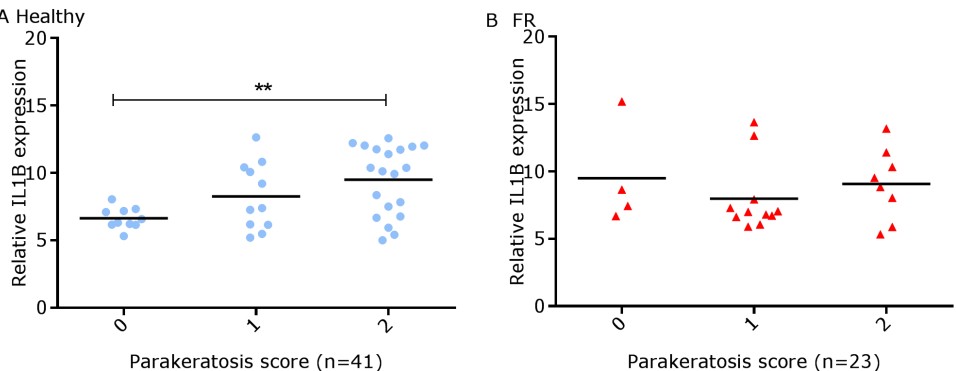

**Figure 7  Comparison between IL-1β mRNA expression and parakeratosis score in ovine interdigital skin.** Correlation between IL-1β mRNA expression and parakeratosis score calculated from Haematoxylin and eosin stained samples of skin/hoof interface using parallel sections from the same interdigital space. $n = 41$ clinically healthy (A) and 23 footrot (B) samples. Statistical analysis: Pearson correlation **$p < 0.01$.

epidermis and 17 dermis). Inflammatory scores increased as virulent *D. nodosus* load increased (Fig. 9). In the epidermis, the virulent *D. nodosus* load was significantly higher ($p \leq 0.0001$) in tissues with an inflammatory score of 3 compared to scores 1 and 2 respectively (Fig. 9B, Fig. S1).

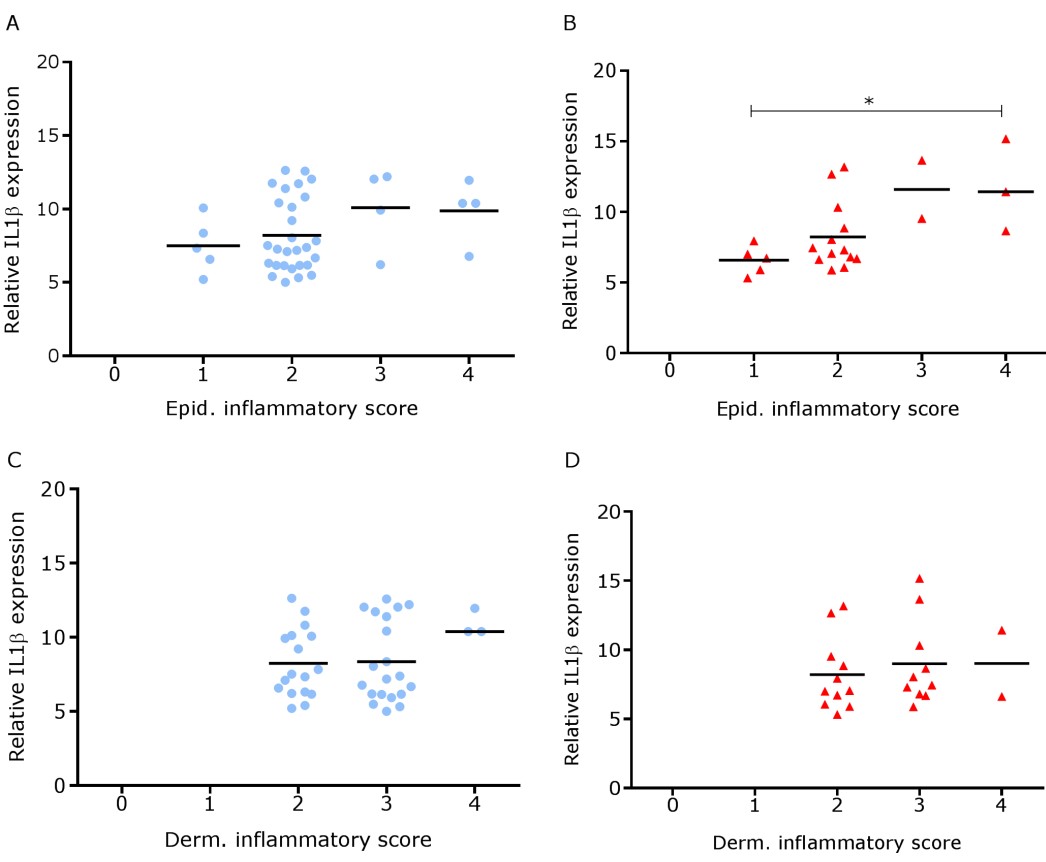

**Figure 8** **Comparison between IL-1β mRNA expression and epidermal and dermal inflammatory score in ovine interdigital skin.** Correlation between IL-1β mRNA expression and inflammatory score calculated from Haematoxylin and eosin stained samples of skin/hoof interface using parallel sections from the same interdigital space. $n = 41$ clinically healthy (A) and 23 footrot (B) from the epidermis and 40 clinically healthy (C) and 23 footrot (D) from the dermis. Statistical analysis: Pearson correlation *$P <$ 0.05.

## Bacterial localisation in ovine interdigital skin

Eubacterial *D. nodosus* and *F. necrophorum* DNA was quantified across the entire skin depth. Eubacterial load in healthy samples ($n = 5$) was similar throughout the tissue depths but peaked at depth of 201–400 μm (Fig. 10). In footrot samples ($n = 4$) eubacterial load progressively decreased from the outermost skin surface to a depth of 1,000 μm, eubacterial load was then similar from 1,200–2,200 μm.

In footrot samples, *D. nodosus* was quantified across depths of 601–2,200 μm, whereas it was detected throughout all levels of the healthy samples. *F. necrophorum* in healthy samples was detected in deeper skin tissues (601–2,200 μm) but was present throughout every levels in footrot samples (Fig. 10). The distribution pattern of eubacterial load in ovine interdigital skin samples were investigated with respect to depth of skin sections in the context of depth of follicles in skin. A relatively low eubacterial load of range of 0.4–19.6 pg/section was observed across healthy samples. In comparison, footrot samples eubacterial load was higher in four out of five samples with peak values of 1,218 pg/section

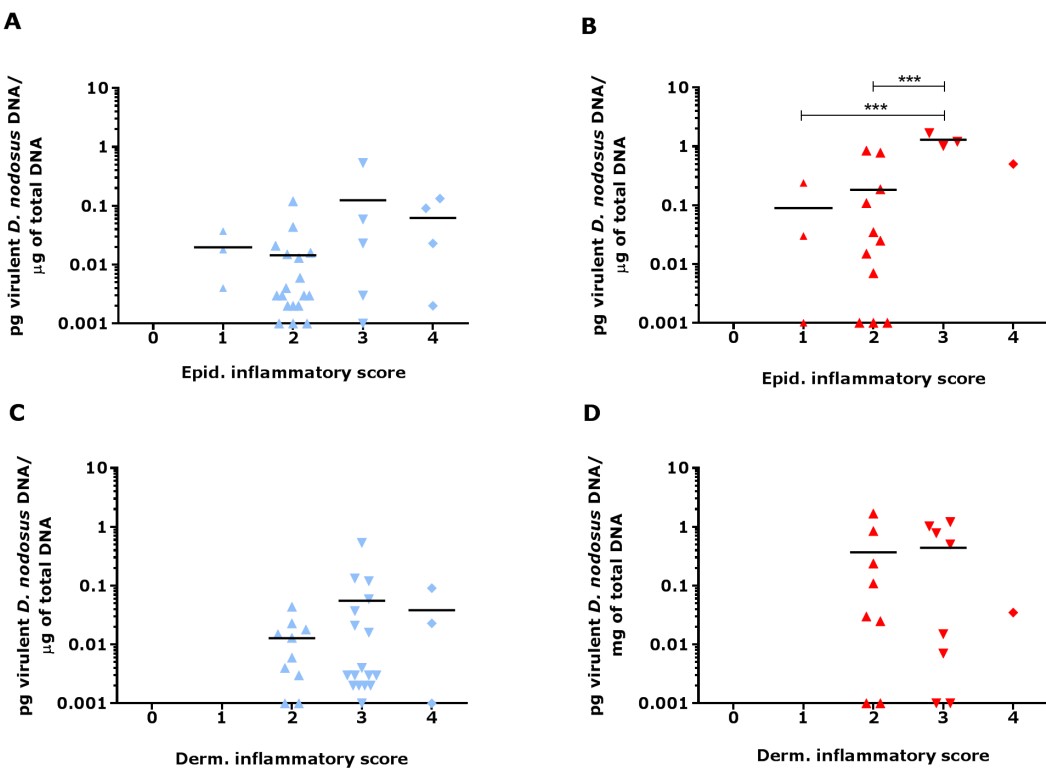

**Figure 9 Virulent *D. nodosus* DNA levels in comparison to inflammatory scores in epidermal and dermal ovine interdigital skin.** Correlation between epidermal and dermal inflammatory scores from healthy ($n = 30$, A and C, respectively) and footrot samples ($n = 18$ epidermis (B) and 17 dermis (D)) in comparison to *D. nodosus* levels. Statistical analysis: Pearson correlation ***$P < 0.0001$.

observed in the superficial skin depth (≤500 μm; Fig. 10). Localisation of eubacterial load did not extend beyond follicular depths (2,119–2,710 μm) in healthy samples while in footrot samples, eubacterial load extended beyond follicular depths (1,460–2,699 μm; Fig. 10, Fig. S2).

## DISCUSSION

In this study, histopathological features of the ovine interdigital skin were analysed and compared between healthy and footrot affected feet using a novel scoring system. In addition, these features were compared to levels of IL-1β and virulent *D. nodosus*. Of note in this study was the severity of inflammatory cell infiltration that was found to be similar between healthy and footrot affected. Inflammation in healthy samples may have been caused by unfavourable ground conditions of the pasture due to wet weather. Under those condition previous studies have also noted an increase in increase the prevalence of footrot and interdigital dermatitis (*Beveridge, 1941*; *Emery, Stewart & Clark, 1984*; *Graham & Egerton, 1968*; *Wassink et al., 2003*; *Wassink et al., 2004*). Alternatively, inflammatory cell infiltrations could be an indication of subclinical disease not yet progressed to visible signs of ID and footrot.

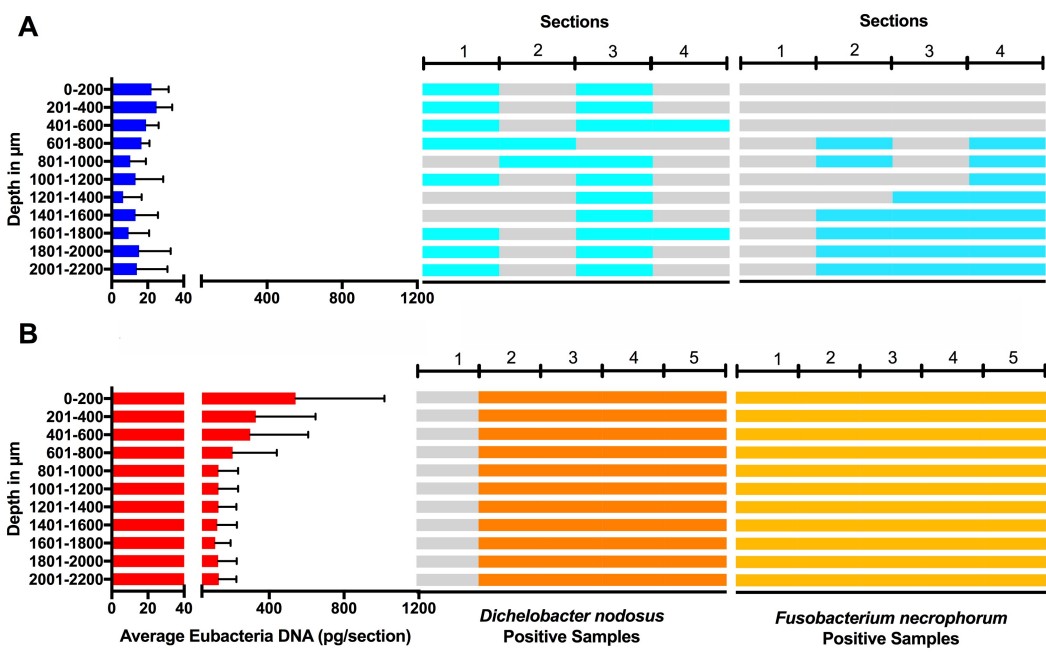

**Figure 10 Bacterial localisation throughout skin layers.** Total DNA was extracted from sequential 40 μm horizontal sections of 6mm punch biopsies of from four healthy (A) and five footrot (B) affected tissue samples. Quantitative PCR was used to enumerate total bacterial DNA end detect the presence or absence of *D. nodosus* and *F. necrophorum*. Sections were grouped in 200 μm bins.

Previous studies have qualitatively described mild inflammatory cell infiltration in healthy ovine feet (*Angell et al., 2015*; *Davenport et al., 2014*). To the best of our knowledge, this is the first study to quantitatively examine inflammatory cells in the ovine feet using a new scoring system. Similar investigations in cattle also found inflammatory cell infiltration in apparently healthy hooves (*Mendes et al., 2013*; *Tarlton et al., 2002*). However, Tarlton and colleagues attributed the inflammatory response observed to the changes in collagen expression and keratinization of epidermal laminae caused by intensive breeding regimes during the peripartum period (*Tarlton et al., 2002*). Interestingly, bovine digital dermatitis, which is driven by inflammatory response in its pathogenesis, is also characterised by inflammatory cell infiltration (*Mendes et al., 2013*; *Refaai et al., 2013*).

Histological lesions including cell ballooning, parakeratosis and micro-abscesses were similarly observed in both conditions and in areas associated with pathology including inflammation in the skin (*Refaai et al., 2013*). Cellular ballooning in the epidermis is an indication of host cell response to stimuli and may be preceded by keratinocytes hyper-proliferating to remove pathogen infected areas (sloughing) (*Edwards, Dymock & Jenkinson, 2003*). Hyper-proliferation of keratinocytes without a corresponding rate of differentiation results in parakeratosis. This has been shown to occur in low grade inflammation caused by microbial presence such as *F. necrophorum* (*Egerton, Roberts & Parsonso, 1969*). Parakeratosis may trigger the release of pro-inflammatory cytokines such as IL-1β (*Chang et al., 1992*) which in-turn mediates the recruitment of inflammatory cells.

This may explain the higher levels of IL-1β expression associated with diffuse parakeratosis in this study.

As expected, higher IL-1β expression levels corresponded with high epidermal inflammatory score grade (score 4) when compared to lower score grade (score 1) in footrot but not in healthy feet. This suggests that IL-1β expression level may be directly proportional to the severity of inflammatory cell infiltrates in the epidermis. To investigate this further, future studies could include immunohistochemical visualization of IL-1β protein in relation to cell infiltration or other lesions. Similarly, significantly increased virulent *D. nodosus* loads tend to correspond with higher epidermal inflammatory score grades (scores 2 and 3) when compared to lower grade (score 1) in footrot but not healthy samples. This suggests that the severity of inflammatory response in footrot is dependent on the abundance of virulent *D. nodosus*. This also confirms previous studies based on severity and bacteria load (*Maboni et al., 2017a*; *Maboni et al., 2016*). The initiation of footrot is mostly associated with virulent *D. nodosus* due to its ability to degrade host extracellular matrix, a trait conferred by the presence of the acidic protease AprV2. It is different from a second benign phenotype which possess the acidic protease AprB2 (*Kennan et al., 2010*). Interestingly, not all cases of virulent *D. nodosus* correlate to severe clinical manifestations since it has been reported in apparently healthy feet without signs of ID or footrot (*Maboni et al., 2016*; *Moore et al., 2005*; *Stauble et al., 2014*). In the UK, virulent *D. nodosus* has been reported as the predominant strain (*Maboni et al., 2016*; *Moore et al., 2005*).

Studies using whole biopsies reported eubacteria presence in healthy ovine interdigital skin (*Calvo-Bado et al., 2011*; *Maboni et al., 2016*; *Witcomb et al., 2015*). Interestingly, we detected eubacterial DNA down to depths of 3 mm in healthy samples. A previous study of healthy human skin also demonstrated eubacterial DNA in deep facial and palm skin tissues (5 mm), with localisation including hair follicles and also dermal stroma (*Nakatsuji et al., 2013*).

As expected, eubacterial load was significantly higher in the outermost layer of footrot skin ($\leq 200 \ \mu$m) in comparison to the same depth in healthy samples. Other studies on a human skin model have also reported higher bacteria abundance in the superficial layers of human skin (*Lange-Asschenfeldt et al., 2011*; *Roeckl & Mueller, 1959*). Detection of bacteria DNA in deep layers of healthy skin may indicate that the skin is not an impervious barrier as previously thought (*Nakatsuji et al., 2013*).

Previous studies based on Giemsa stain and fluorescent *in situ* hybridisation (FISH) have suggested *D. nodosus* was primarily localised in superficial epidermis of ovine interdigital skin (*Egerton, Roberts & Parsonso, 1969*; *Witcomb et al., 2015*). However, Witcomb and colleagues reported a single cell of *D. nodosus* in the dermis of footrot infected ovine skin (*Witcomb et al., 2015*). In contrast to FISH, qPCR technique is able to amplify target DNA, thereby detecting low levels of *D. nodosus*.

Hair follicles have been reported to serve as a bacterial reservoir and hence, possible routes of bacterial entry into healthy skin (*Montes & Wilborn, 1970*). The analysis of sequential transverse sections in our study showed that eubacterial load corresponded to follicular depth in healthy samples but extended beyond follicular depth in footrot samples. This suggests that hair follicles may play a role in eubacteria localisation in intact healthy

skin. In contrast, in a similar polymicrobial disease of bovine feet, digital dermatitis, no association was found between treponemes and hair follicles (*Evans et al., 2009*).

## CONCLUSIONS

Data presented in this study showed for the first time that there are no consistent differences in the level and range of histological lesions examined between healthy and footrot affected feet. Interestingly, the novel inflammatory cell infiltration scoring system developed and validated in this study mirrored the pro-inflammatory cytokine IL-1β and confirmed an association between severity of inflammatory response and increased virulent *D. nodosus* load.

## ACKNOWLEDGEMENTS

The authors thank Aziza Alibhai for her technical assistance with the histology as well as the abattoir staff for assistance with sample collection.

### Funding

This study was funded by the Governments of Nigeria (TETFUND) and Brazil (CAPES) and The School of Veterinary Medicine and Science, University of Nottingham. The funders had no role in study design, data collection and analysis, decision to publish, or preparation of the manuscript.

### Grant Disclosures

The following grant information was disclosed by the authors:
Governments of Nigeria (TETFUND) and Brazil (CAPES).
School of Veterinary Medicine and Science, University of Nottingham.

### Competing Interests

The authors declare there are no competing interests.

### Author Contributions

- Michael Agbaje performed the experiments, analyzed the data, prepared figures and/or tables, authored or reviewed drafts of the paper, approved the final draft.
- Catrin S. Rutland conceived and designed the experiments, performed the experiments, analyzed the data, contributed reagents/materials/analysis tools, prepared figures and/or tables, authored or reviewed drafts of the paper, approved the final draft.
- Grazieli Maboni, Adam Blanchard, Melissa Bexon and Ceri Stewart performed the experiments, analyzed the data, prepared figures and/or tables, approved the final draft.
- Michael A. Jones conceived and designed the experiments, performed the experiments, analyzed the data, contributed reagents/materials/analysis tools, prepared figures and/or tables, authored or reviewed drafts of the paper, approved the final draft, funding.
- Sabine Totemeyer conceived and designed the experiments, performed the experiments, analyzed the data, contributed reagents/materials/analysis tools, prepared figures and/or

tables, authored or reviewed drafts of the paper, approved the final draft, funding and ethics.

## Animal Ethics

The following information was supplied relating to ethical approvals (i.e., approving body and any reference numbers):

Ethical approval was obtained from the School of Veterinary Medicine and Science Ethics committee, University of Nottingham, UK (ethical approval number: 796 130128).

## Data Availability

The raw data are provided as Supplemental Files.

## Supplemental Information

Supplemental information for this article can be found online at http://dx.doi.org/10.7717/ peerj.5097#supplemental-information.

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
