# Peer review of "Novel inflammatory cell infiltration scoring system to investigate healthy and footrot affected ovine interdigital skin"

_PeerJ, doi:10.7717/peerj.5097_

## Round 0.1 · original submission · Minor Revisions

We apologize for delay in revisions. Please, find attached reviewer comments that must be addressed throughout the manuscript in order to improve the experimental design setting and validity of the findings.

Reviewer 1 ·

Basic reporting

This manuscript reports a histopathological comparison of interdigital skin lesions of healthy and footrot affected sheep.
The manuscript correctly matches with the scope of the journal. It is correctly described and English language is properly used. The general structure is according to the standards of the journal and is clear for the reader.
Figures are of great quality and represent the main findings that the authors stated in the manuscript.
Minor editing points of criticism have been detected in lines:
236 to 238: necrophorum (Egerton et al. 1969). Parakeratosis may trigger the release of pro-inflammatory cytokines such as IL-1β (Chang et al. 1992) which in-turn MEDIATES the recruitment of inflammatory cells.
278. 278 polymicrobial disease of bovine feet, digital DERMATITIS, no association was found between
358: Total DNA was extracted from sequential 40 µm horizontal sections of 6mm punch biopsies of from…
360: enumerate total bacterial DNA AND detect…

Experimental design

The authors should reconsider the title of the manuscript because it doesn’t reflect the findings of the study. What is novel in the research is the “inflammatory cell infiltration scoring system”. The rest of the study shows that the histopathology findings are quite similar both in healthy and affected sheep. Nevertheless, the title can also include all the descriptive work that has been done.
About this second affirmation, is missing in the discussion part why the histopathological findings are similar in healthy and footrot affected animals. Are the clinical findings correctly diagnosed?. Are we underdiagnosing footrot animals?. The animals are suffering an subclinical disease?. Please, reconsider all this topics and discus in the manuscript.
The investigation presented is in agreement with the scope of the journal. The methodology is rigorously described and the experiments are reproducible.

Validity of the findings

Although results are presented whose findings are inconclusive, it is considered that it can help the scientific community to continue with this study in a rigorous manner.
The data presented is strong and the statistical analysis is well applied.
Conclusions are well stated.

Reviewer 2 ·

Basic reporting

The paper by Agbaje et al. studies by histopathological methods natural cases of ovine footrot together with etiological and some immunological approaches in relation with the lesions. Not too much work has been done on this particular disease, especially in the pathogenesis of the lesions. In this sense, the paper would contribute
This is a well-written and clearly presented paper that includes representative and high-quality figures that clarify the histological findings reported in the paper. The literature review is accurate and up-to-date.

Experimental design

There are some concerns concerning this paper:
- The aim of the study is not clear at all, more than performing a grading of histological lesions.
- It is not clear enough the aim prosecuted by the study of IL1beta mRNA expression in association with the lesions. Considering the samples and material, they should have made an attempt to go further into pathogenesis of the lesion, as they try to, when measure IL1betta expression. It could be examined the relationship of its expression with the presence of different cell populations, location of lesions, etc.
- Why the authors have chosen this particular cytokine and no other, including inflammatory mediators, that would have given more information on the development of the lesions in relation with the host immune response?
- The methods for the histological studies are classical and well-presented. In relation to HE staining, why the authors have decided to treat the samples with acetic spirits and ammoniated water? Which are the advantages respect to conventional HE staining?
- Methods for bacteriological and statistical analysis seem to be appropriate.
- Which criteria have been taken into account for the gross classification of the samples into healthy and footrot affected? Please, state in the material and methods section.

Validity of the findings

The main lack of this work is the scarce novelty of the scoring system the authors have proposed. They classify the inflammatory infiltrate according to the intensity, following a very classical pathological scoring system. Moreover, there are other facts that they also consider, but separately. Putting all together, they do not find any difference between healthy and diseased samples. All the histological findings considered are classical and have been previously well described and reported, as is mentioned in the paper.
The authors should have made a more ambitious attempt of classification putting considering all the changes together and giving a particular score to each animal, and try to find a cut-off value to discriminate between healthy and footrot affected animals. This would help to the diagnosis and future studies on pathogenesis.
Regarding the role of IL1beta, the authors have just measured the mRNA expression of this cytokine. It would be of interest to assess the protein expression by immunohistochemistry, in relation with the lesions.
The study made on the presence of different bacteria in association do not provide new or relevant findings. The role of the different bacteria have been elucidated in others papers of the group. The presence in the various depths of the lesion is the only new finding, but its relevancy is not well-stablished.

---

## Round 0.2 · accepted · Accept

I confirm that the reviewers' comments have been addressed thoroughly.

#